

# Influence of water availability and temperature on estimates of microbial extracellular enzyme activity

Enrique J. Gomez, Jose A. Delgado and Juan M. Gonzalez

Instituto de Recursos Naturales y Agrobiología, Consejo Superior de Investigaciones Científicas, IRNAS-CSIC, Sevilla, Spain

## ABSTRACT

Soils are highly heterogeneous and support highly diverse microbial communities. Microbial extracellular enzymes breakdown complex polymers into small assimilable molecules representing the limiting step of soil organic matter mineralization. This process occurs on to soil particles although currently it is typically estimated in laboratory aqueous solutions. Herein, estimates of microbial extracellular enzyme activity were obtained over a broad range of temperatures and water availabilities frequently observed at soil upper layers. A *Pseudomonas* strain presented optimum extracellular enzyme activities at high water activity whereas a desiccation resistant bacterium (*Deinococcus*) and a soil thermophilic isolate (*Parageobacillus*) showed optimum extracellular enzyme activity under dried (i.e., water activities ranging 0.5–0.8) rather that wet conditions. Different unamended soils presented a distinctive response of extracellular enzyme activity as a function of temperature and water availability. This study presents a procedure to obtain realistic estimates of microbial extracellular enzyme activity under natural soil conditions of extreme water availability and temperature. Improving estimates of microbial extracellular enzyme activity contribute to better understand the role of microorganisms in soils.

## INTRODUCTION

Soil microorganisms play a critical role in the cycling of elements maintaining soil health. However, little is known about how microbial activity is influenced by soil features and the consequences of these interactions on local and global scales. This is due, for example, to the complexity of the soil environment and the huge functional and taxonomical diversity observed in the microbial world (*Lupatini et al., 2017*; *Whitman, Coleman & Wiebe, 1998*) which are poorly understood (*Meyer et al., 2018*).

Soil biogeochemical cycling of elements is governed to a large extent by the activity of microorganisms (*Conant et al., 2011*). Nutrients and their availability, including organic matter, are the result of a balance between the biotic (their use as source for growth and maintenance of organisms, both microbes and macro-organisms) and abiotic (including complex chemical interactions) components. In this respect, the stock of carbon in soil is larger than the atmospheric stock and carbon accumulation and mineralization

Corresponding author
Juan M. Gonzalez,
jmgrau@irnase.csic.es

represent major factors affecting the global soil-atmosphere carbon balance which influences global warming (*Davidson & Janssens, 2006*; *Intergovernmental Panel on Climate Change, 2020*).

Soil organic carbon decomposition is mainly performed by microorganisms (*Conant et al., 2011*; *Whitman, Coleman & Wiebe, 1998*). To process soil polymers and high molecular weight compounds, microorganisms need to break them down into monomers or assimilable, smaller subunits that can be taken up for further metabolic processing (*Asmar, Eiland & Nielsen, 1994*; *Madigan, Martinko & Parker, 2003*). This first step of decomposition is carried out by microbial extracellular enzymes and their production is regulated by the microbial communities as a function of the availability of different organic substrates (*Velasco-Ayuso et al., 2011*) and the environment (*Santana, Gonzalez & Garbeva, 2015*). These extracellular enzymes represent the rate-limiting step for the biological consumption and mineralization of soil organic matter (*Chróst, 1992*; *Conant et al., 2011*; *Gonzalez, Portillo & Piñeiro-Vidal, 2015*) and their activities are expected to be ruled by environmental conditions (i.e., temperature, moisture, pH).

Upper soil layers are exposed to broad variations of environmental conditions, such as temperature and moisture. For instance, arid, semi-arid and desert soils frequently reach high temperatures. Some reports mentioned common temperatures in the range of 50–70 °C in temperate soils from medium latitudes (*Gonzalez, Portillo & Piñeiro-Vidal, 2015*) and values above 90 °C in deserts (*McCalley & Sparks, 2009*). For example, soils from Northern and Southern Spain are exposed to clearly different climate conditions (i.e., Atlantic vs Mediterranean climates, respectively). Northern Spain is rainy with mild temperatures while Southern Spain show higher temperature and dry periods with a more pronounced risk of aridity. Besides, the importance and ubiquity of thermophilic microorganisms in soils have been reported (*Marchant et al., 2002*; *Portillo, Santana & Gonzalez, 2012*) including cold areas at high latitudes (*Wong et al., 2015*). Recently, the effect of high temperatures has been associated to a decrease of soil organic content (*Biederman et al., 2016*; *Bragazza et al., 2016*; *Cheng et al., 2017*; *Santana, Gonzalez & Garbeva, 2015*) and so activity of thermophilic microorganisms and climate change appear to be linked (*Santana, Gonzalez & Garbeva, 2015*). About the influence of wet/drying periods, there are numerous reports showing an enhancement of microbial and enzymatic activities in soils after rain or during wetting cycles in contrast to lower activities during dried intervals (*Conant et al., 2011*; *Manzoni, Schimel & Porporato, 2012*). Current knowledge suggests the existence of highly dynamic microbial communities and extracellular enzyme activities in soils although the detailed effects of temperature and water availability on microorganisms and their extracellular enzymes are poorly understood (*Conant et al., 2011*; *Gonzalez, Portillo & Piñeiro-Vidal, 2015*; *Manzoni, Schimel & Porporato, 2012*; *Santana, Gonzalez & Garbeva, 2015*; *Wallenstein & Weintraub, 2008*).

High temperatures tend to be accompanied with low water content in soils (*Biederman et al., 2016*). However, most soil enzymatic assays to measure enzyme activities are performed in solution at or below 30 °C (*Craine et al., 2010*; *Fierer et al., 2006*; *Townsend, Vitousek & Holland, 1992*) which is not necessarily representative of the actual conditions found in soil upper layers and, above all, those with scarce plant coverage during the

summer season. Currently, there is interest to approximate field conditions to perform extracellular enzyme assays in soil samples (*Wallenstein & Weintraub, 2008*). For instance, peaks of maximum enzymatic activity have been recently observed at high temperatures in all soil samples (*Gonzalez, Portillo & Piñeiro-Vidal, 2015*) from a wide range of latitudes and soil types. Besides, enzymatic assays to approach hydrolytic enzyme activity in soils are typically carried out in solution. Unlike it, upper soil layers can get dry. Decreasing water availability in soils results in desiccation events leading to major changes in the microbial communities and certainly in their enzyme activities catalyzing soil organic matter decomposition (*Biederman et al., 2016*; *Cheng et al., 2017*; *Conant et al., 2011*; *Tecon & Or, 2017*). The combined effects of high temperatures and low water availability on soil microbial extracellular enzyme activity remain to be understood both for soil mesophilic and thermophilic bacteria.

Water availability is generally measured through the parameter water activity ($a_w$) (*Grant, 2004*). Water activity represents the water available for microorganisms and it is defined as the partial vapor pressure of water in a sample divided by the partial vapor pressure of pure water. The $a_w$ can get values from 1 (in a water saturated sample) to a theoretical value of 0 (in a sample with no available water). A few attempts have been reported in the laboratory to analyze microbial enzymatic activity and water availability (*Borowik & Wyszkowska, 2016*; *Stark & Firestone, 1995*). Most studies on this issue have been carried out on the industrial field (*Lyer & Ananthanarayan, 2008*; *Soares, Teixeira & Baptista, 2003*; *Torres & Castro, 2004*). It is assumed that while the active center conserves water above a minimum threshold, enzyme activity continues but below that level it will be drastically reduced. Also, maximum activity might not necessarily occur at excess of water. Generally, the procedures for these measurements involve the use of salts, sugars and/or chemicals (i.e., organic solvents) to control water activity during experimental analyses over relatively long time periods (*Hudson, Eppler & Clark, 2005*; *Maurice et al., 2011*). However, the addition of most of these water activity-limiting chemical mixtures might alter, or might not be representative of, the behavior of microorganisms and their enzymes in soils. For example, supplementation with salts or organic solvents can affect protein structure and so enzyme and microbial activities (*Borowik & Wyszkowska, 2016*; *Hudson, Eppler & Clark, 2005*; *Lee & Dordick, 2002*; *Oren, 2010*). Soil drying processes are not necessarily homogenous among soil particles and their cavities can form heterogenous microniches (*Tecon & Or, 2017*; *Vos et al., 2013*).

The influence of water availability (under low water activity conditions) on natural soil microorganisms and their EEA remains to be clearly understood. Due to the non-linear relationship between water activity and moisture (*Mathlouthi, 2001*), soil moisture (by weight or volume) values generally reported correspond to measurements at high water availability values (moisture ≥10% generally corresponds to $a_w$ above 0.8) thus excluding information from low water availability conditions from experimentation (*Mathlouthi, 2001*; *Grant, 2004*; *Steinweg, Dukes & Wallenstein, 2012*; *Moxley et al., 2019*). Similarly, enzyme activity in soil has been measured at relatively elevated water potential values (generally >−4 Mpa) (*Stark & Firestone, 1995*; *Steinweg, Dukes & Wallenstein, 2012*) also representing high water activity values ($a_w > 0.9$). Thus, the ideal approach to

assess enzyme activity in soils should avoid or minimize the addition of supplements, be performed within a reasonable short timeframe, and approach low water activity conditions.

Currently, soil extracellular enzyme activities are mainly determined in laboratory aqueous solutions. However, soils are highly heterogeneous environments formed by a conglomerate of particles where water is frequently limited due to the meteorological conditions directly affecting upper soil layers. Drying soils have been reported to decrease or even completely suppress enzyme activity (*Allison & Treseder, 2008*; *Duran & Esposito, 2000*). Thus, it is required to approach microbial extracellular enzyme activity estimates with more realistic procedures in order to evaluate microbial extracellular enzyme activity in soils under a broad range of temperature and water availability conditions.

This study aims (i) to propose a new method to estimate microbial extracellular enzyme activity at different temperatures and water activities within the range of values detected in some soils (e.g., including arid systems); (ii) to gain information of the behavior of extracellular enzyme activities from specific bacteria in sterilized soils so that we can understand the influence of temperature and water availability on their capacity to hydrolyze complex organic nutrients; and (iii) to compare the activities of extracellular enzymes from two Spanish soils exposed to different environmental conditions emphasizing the specific properties of soils exposed to aridity.

## MATERIALS AND METHODS

### Experimental design and sampling sites

Extracellular enzyme activity was determined applying an assay protocol (see below) designed to reproduce the potential conditions of temperature and dryness that can be found at the upper layers of typical soils. Two temperatures, 20 °C and 60 °C, were selected which represented common conditions of the extracellular enzyme activity from mesophilic (*Fierer et al., 2006*; *Townsend, Vitousek & Holland, 1992*) and thermophilic (*Gonzalez, Portillo & Piñeiro-Vidal, 2015*) microorganisms. Water availability was determined by measuring water activity ($a_w$) using a Rotronic water activity probe HC2-AW (Rotronic AG, Bassersdorf, Switzerland) at the incubation temperature. Extracellular enzyme activity assays were carried out over different water activities, decreasing values from 1 down to the water activity resulting in zero or near-zero enzyme activity.

Two types of experiments were carried out. Measurements using either unsupplemented natural soils or sterilized soils supplemented with bacterial cultures were analyzed. In order to obtain an image of the behavior of the enzymes from specific bacteria, we tested the extracellular enzymatic activity of three bacterial species isolated from soils: *Pseudomonas aeruginosa*, *Deinococcus radiodurans* and *Parageobacillus thermoglucosidasius*. Table 1 lists the strains used in this study. Bacterial cultures were grown in nutrient broth (Cat. No. 234000; BD Difco, Franklin Lakes, NJ, USA) at their optimum growth temperature (Table 1). Bacterial cells from these species at late exponential growth phase were supplemented to sterilized soil samples from a Southern Spain soil (see below) as a support to simulate potential soil conditions of different water content during the enzyme assays.

**Table 1 Properties for the bacterial species used in this study.**

| Species | Strain | Accession No.[1] | Optimum growth temperature | Classification by temperature | Phylum |
|---|---|---|---|---|---|
| *Pseudomonas aeruginosa* | PAO1 | CECT4122 | 35 °C | Mesophile | Proteobacteria |
| *Deinococcus radiodurans* | R1 | CECT833 | 28 °C | Mesophile | Deinococcus/Thermus group |
| *Parageobacillus thermoglucosidasius* | 23.6 | CECT9776 | 60 °C | Thermophile | Firmicutes |

Note:
[1] CECT, Colección Española de Cultivos Tipo, Spanish Type Culture Collection.

**Table 2 Some characteristics of the two natural soils analyzed in this study.**

| Location | Municipality | Coordinates | Temperature (annual mean; °C) | Precipitation (annual mean; mm) | Climate type[1] | Soil type | Sand/Silt/Clay content (%) | pH | Organic matter (%) |
|---|---|---|---|---|---|---|---|---|---|
| Southern Spain | Coria del Rio, Sevilla | N 37° 17.027′ W 006° 3.973′ | 18.4 | 572 | Csa | Sandy loam | 69.4/27.8/2.8 | 7.4 | 7.0 |
| Northern Spain | Benasque, Huesca | N 42° 40.922′ E 000° 38.108′ | 8.2 | 1,013 | Cfb | Silt | 12.6/83.3/4.1 | 6.5 | 12.2 |

Note:
[1] Köppen–Geiger climate classification.

The amended soil was previously sterilized by autoclaving at 121 °C for 30 min during five consecutive days which removed all detectable enzyme activity in these aliquots.

The assays carried out on natural soil samples were processed unsupplemented and without treatments. The soil samples were collected from distant locations and corresponded to a sandy loam soil (Coria del Rio, Sevilla, Southern Spain; 37° 17.027′ N, 006° 3.973′ W; Dry summer, Mediterranean climate (Csa)) and a silt soil (Benasque, Huesca, Northern Spain; 42° 40.922′ N 000° 38.108′ E; Oceanic, Northern Europe climate (Cfb)) (Table 2). These two soils were selected as examples of soils from different climates (humid vs dry, cold vs hot) to compare the influence on extracellular enzyme activity as a function of temperature and water activity. It is important to underline that the temperature and water activity to what soil microorganisms and their enzymes are exposed would induce a broad range of responses (*Alster, Weller & Von Fischer, 2018*).

## Enzyme assays

Most assays to evaluate the microbial extracellular enzyme activity in different environments are performed in solution. Unlike previous studies, we compared the activity determined in solution and following the procedure detailed below which accounts for environmental factors because they are carried out on the soil particles and at a specific water activity and temperature.

Extracellular glucosidase, protease and phosphatase activities were assayed as examples of extracellular enzyme activity in soils and represent activities commonly assayed in environmental enzyme activity surveys. These enzymes represent key steps of the process of depolymerization of high-molecular weight organic matter in soils in relationship to the C biogeochemical cycle and, specifically the protease and phosphatase, to the N and P cycles,

respectively. Reactions were buffered by phosphate buffer (0.2 M, pH 7) for protease and glucosidase assays, and PIPES buffer (2 mM; piperazine-$N,N'$-bis[2-ethanesulfonic acid]; pH 7) for the phosphatase assay. Buffer solutions were adjusted for pH at the temperature to be used. Enzyme assays were based on the use of the fluorogenic substrates L-leucine-7-amido-4-methylcoumarin hydrochloride (AMC) for protease activity, Methylumbelliferyl β-glucopyranoside (MUG) for glucosidase activity and Methylumbelliferyl phosphate (MUP) for phosphatase activity. These substrates showed stability under the assayed temperature conditions as they have been previously evaluated up to 100 °C (*Gonzalez, Portillo & Piñeiro-Vidal, 2015*). Figure 1 shows a scheme of the assays comparing a standard, in solution assay and the newly proposed assay on soil samples which allows the evaluation of dryness on soil extracellular enzyme activities.

## Enzyme assays under dried conditions

Enzymatic assays (Fig. 1) were performed by adding a buffered solution containing the corresponding fluorogenic substrate (0.1 mM, final concentration; AMC, MUG and MUP) (*Gonzalez, Portillo & Piñeiro-Vidal, 2015*) to natural soil (2 mg aliquots) or the sterilized soil supplemented with bacterial species. Until this point samples and solutions were maintained on ice and once the fluorogenic substrates were added, the soil mixtures were frozen at −80 °C to minimize substrate degradation and modification of sample conditions during handling. Samples aimed to perform enzyme activity measurements at reduced water activities aliquots were freeze dried to reduce the water content in the soil mixture and reach the required water activity. The final water activity was determined as detailed above using a Rotronic water activity probe. Samples were only freeze-dried a single period and those at a water activity close to the required water content were used for the assay. If the resulting water activity at a given soil subsample was distant from the expected value that subsample was not processed further. Reactions at different temperatures and water activities were carried out in triplicate. Once the required water activity is obtained, the aliquots were incubated in a sealed container at the required temperatures (20 °C or 60 °C) for different time periods. Different aliquots were placed in different tubes so that for each time point three replicates were extracted from incubation. Incubation times were below 10 min because around this time the kinetic curve leveled-off. That time period was sufficient to estimate the linear slope of fluorescence vs incubation time in the studied cases. The enzyme assays for natural samples were incubated at 20 °C and 60 °C. Enzyme assays for bacterial cultures were incubated at the strain optimum growth temperature. Time zero was considered when the reaction reached the desired incubation temperature. After the incubation period, the reactions were stopped by adding ethanol (*Stemmer, 2004*) and the pH was adjusted with ice-cold glycine-NaOH (0.1 M; pH 11) to maximize the fluorescence signal while preserving the undigested fluorogenic substrate and the fluorescent product. The stopped reaction mixture was vortexed and the solution was cleared from soil particles by centrifugation at 5,000×*g* for 5 min (4 °C). Fluorescent measurements were carried out in an Omega fluorometer (BMG LabTech GmbH, Ortenberg, Germany) using the filter sets recommended by the manufacturer (excitation 355 nm; emission

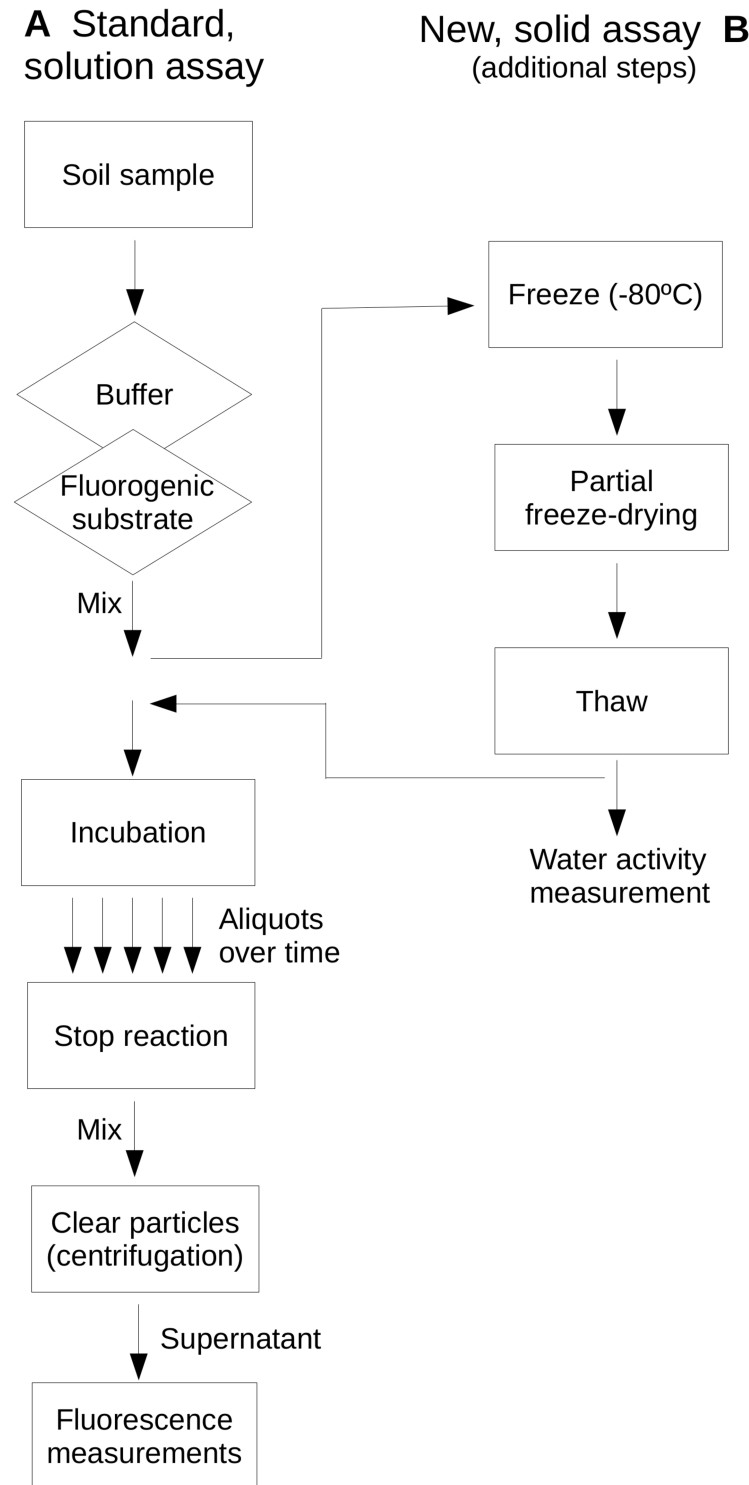

**Figure 1 Comparative scheme of the standard, in-solution assay (A) and the newly proposed, soil-based assay (B) used in this study.** Note that the same protocol applies to both assays with the exception of the additional steps required in (B). A diamond indicates a step where a reagent solution is added; the rectangle indicates other steps.

460 nm). The rate of enzyme activity was estimated by linear regression (Model I, only Y variable is subject to error) (*Sokal & Rohlf, 2012*) as the slope of fluorescence vs incubation time during the linear portion of the curve. Correlation and regression analysis (Model II, both variables are subject to error) between the results from assays following the above protocol and those performed by the classic, in-solution method (see below) where performed according to *Sokal & Rohlf (2012)*.

### In-solution enzyme assays

As a reference, enzymatic assays were also performed following the standard procedure for the estimation of extracellular enzyme activity in soils (*Wallenstein & Weintraub, 2008*). To allow a comparative analysis, in-solution assays followed the protocol described above with natural soils samples but omitting the water reduction step by freeze-drying (Fig. 1).

## RESULTS

### Alternative assay under low water availability

In this study, we propose a novel assay for the determination of extracellular enzyme activity at reduced water activities and different temperatures which are conditions typically encountered at upper soil layers. The required water activities are obtained by freeze drying the reactions which efficiently removed a fraction of the water contained in the reactions. It is important to consider that, in order to minimize the activity of enzymes on the fluorogenic substrates, the reaction mixtures must be frozen. This is also a necessary step in the freeze drying process. Once the required water activity of a reaction mixture is obtained, the incubation reaction will start just when the reaction reaches the required temperature. Time zero aliquots will be the initial fluorescence reading which accounts for potential minimum degradation of fluorogenic substrates previous to the incubation period. Readings from later time periods showed an increase over time until a plateau of maximum fluorescence yield is reached at approximately 10 min incubation. Enzyme activity rates vs substrate concentration estimated by using the proposed procedure followed Michaelis–Menten kinetics (Fig. 2).

The proposed procedure provided data equivalent to those obtained by assays in solution (Fig. 3). The extracellular enzyme activity measurements from two different soil types resulted in a significant relationship (correlation coefficient $r = 0.90$; $n = 12$; $P < 0.001$). The 95% confidence limit of the estimated slope included the 1:1 reference line showing that both assays provided similar results when compared at $a_w$ 1. At the highest measured enzyme activities, the results tended to the largest differences between the in-solution and solid assays, showing slightly higher values in solution. This trend would suggest that substrate diffusion (*Tecon & Or, 2017*) could start to become an issue under high enzyme activity conditions in soils. At $a_w < 1$ comparisons become impossible because the in-solution assay can not be performed (i.e., without the addition of organic solvents or salts that could negatively affect the activity of soil enzymes). Results of fitting a Gaussian curve (Normal distribution) to replicates from in-solution and solid assays

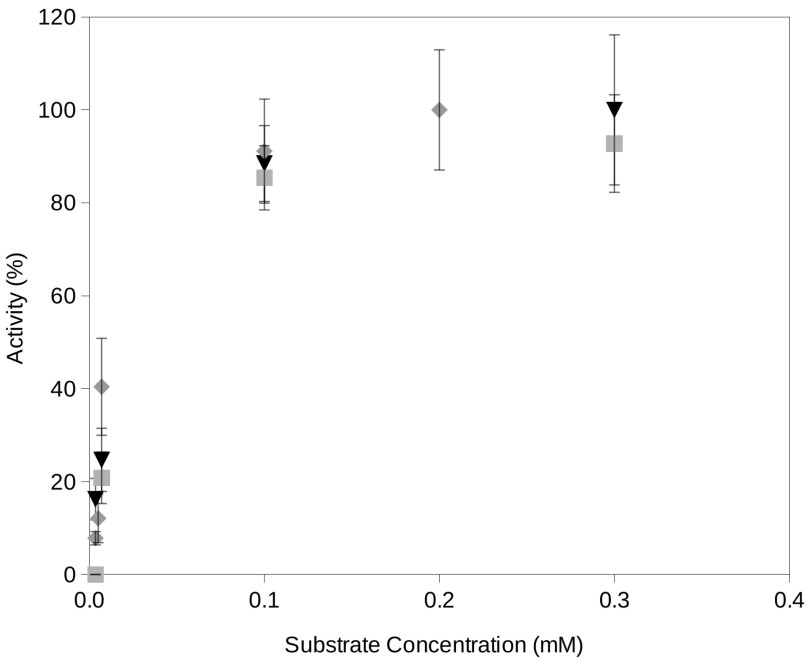

**Figure 2 Examples of the activity rate vs substrate concentration at water activity 0.95 using the proposed soil-based assay on a South Spain soil.** Examples of the activity rate (i.e., fluorescence increase over time) vs substrate concentration at water activity 0.95 using the proposed soil-based assay on the South Spain soil. The results of this method showed a Michaelis–Menten kinetics. Grey square, phosphatase activity; black triangle, protease activity; gray diamond, glucosidase activity.

suggested a larger variability (2-fold) of the solid assays likely due to the heterogeneity typically found in soil particles vs the homogeneity of solutions.

## Extracellular enzyme activity by bacterial isolates

To evaluate the behavior of extracellular enzymes in soils we first tested three bacterial cultures to approach how these enzymes could function under a range of water activity conditions commonly observed at upper soil layers. The results of specific bacterial species should be a necessary step towards attempting to understand the behavior of extracellular enzyme activities in soils. Figure 4 shows the results obtained from the extracellular enzyme activities measured for three cultured species at different water activity values. For instance, the proteobacterium *P. aeruginosa*, a well studied mesophilic strain, showed maximum extracellular enzyme activity at water activity equal to 1. When the water activity was reduced, the extracellular enzyme activity of this bacterial species rapidly decreased so that glucosidase and phosphatase activities dropped to undetectable rates at water activity values of 0.8 (Fig. 4). When the water activity was reduced to 0.8, the protease activity for *P. aeruginosa* remained around 80% (Figs. 4A and 4B) and sharply decreased at lower $a_w$ values. Similar behavior was observed for a laboratory strain, *Escherichia coli* (K12) (data not shown).

*Deinococcus radiodurans* is a model bacterium for the study of desiccation resistance (*Potts, 1994*; *Slade & Radman, 2011*). When the response of extracellular enzyme activity

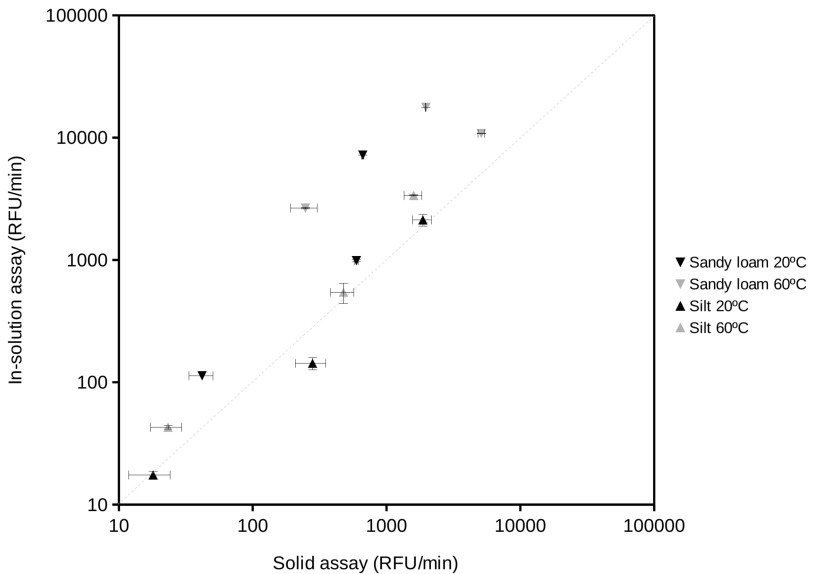

**Figure 3 Comparison of the results obtained using the standard enzyme activity assay (in-solution assay) and the proposed soil-supported activity method at the same substrate concentration, temperature and water activity ($a_w = 1$) conditions.** Reactions at 20 °C and 60 °C were performed to estimate protease, phosphatase and glucosidase activities in two natural soil samples, a sandy loam (Southern Spain) and a silt (Northern Spain) soil. Correlation coefficient between these data was 0.90 ($n = 12$; $P < 0.001$). The dashed straight line indicates the 1:1 correspondence as a reference. Axes are in logarithmic scale. RFU, Relative Fluorescent Units.

by *D. radiodurans* was evaluated as a function of water activity, the results were slightly different depending on which type of enzyme was analyzed (Fig. 4C). *D. radiodurans* glucosidase activity decreased progressively at decreasing water activity. However, its phosphatase activity was mantained constant in the range of water activities from 0.55 to 1 although *D. radiodurans* phosphatase activity drastically dropped to zero at $a_w$ 0.4. The protease activity of this bacterium showed a peak of maximum activity at $a_w$ 0.55 and for $a_w$ 0.35 its activity was sharply reduced to undetectable levels (Fig. 4). Interestingly, *D. radiodurans* protease activity under wet conditions ($a_w = 1$) showed a poor activity rate (30% of its maximum at $a_w$ 0.55).

An example of a thermophilic soil isolate is *Parageobacillus thermoglucosidasius* which is a common representative of the thermophilic Firmicutes found in soils (*Santana, Gonzalez & Garbeva, 2015*). For this isolate, glucosidase and protease activities showed peaks of optimum activity at $a_w$ 0.8 (Fig. 4) decreasing at lower water activities. Its activity was relatively low (20–30% of maximum values) under wet conditions ($a_w = 1$). The *P. thermoglucosidasius* isolate phosphatase activity showed moderate enzymatic activity at high $a_w$ values (above 0.7) and a similar model to its other extracellular enzymes. Nevertheless, its phosphatase activity was observed at $a_w$ 0.6 and sharply decreased to undetectable values at $a_w$ 0.5 (Fig. 4).

## Extracellular enzyme activity in natural soil samples

Soil extracellular enzyme activities were assayed to quantify the enzyme activities of mesophilic (20 °C) and thermophilic (60 °C) microorganisms over a broad range of water

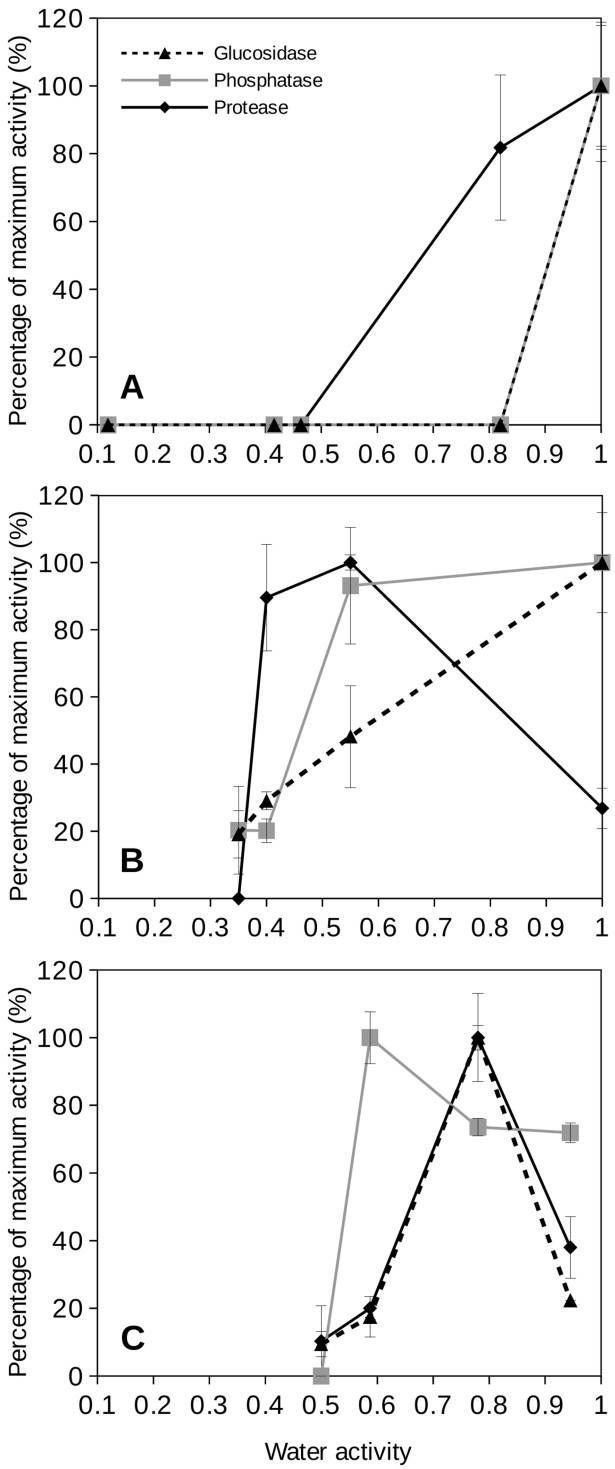

**Figure 4 Relative extracellular enzyme activity as a function of water activity ($a_w$) for three bacterial species.** Relative extracellular enzyme activity as a function of water activity ($a_w$) for three bacterial species: (A) *Pseudomonas aeruginosa* (35 °C); (B) *Deinococcus radiodurans* (28 °C), and (C) *Parageobacillus thermoglucosidasius* (60 °C). Three different extracellular activities were tested: glucosidase (blue triangles), phosphatase (red squares) and protease (green diamonds) activities. Error bars represent one standard deviation. Results are shown as percentages of the maximum activity rate.

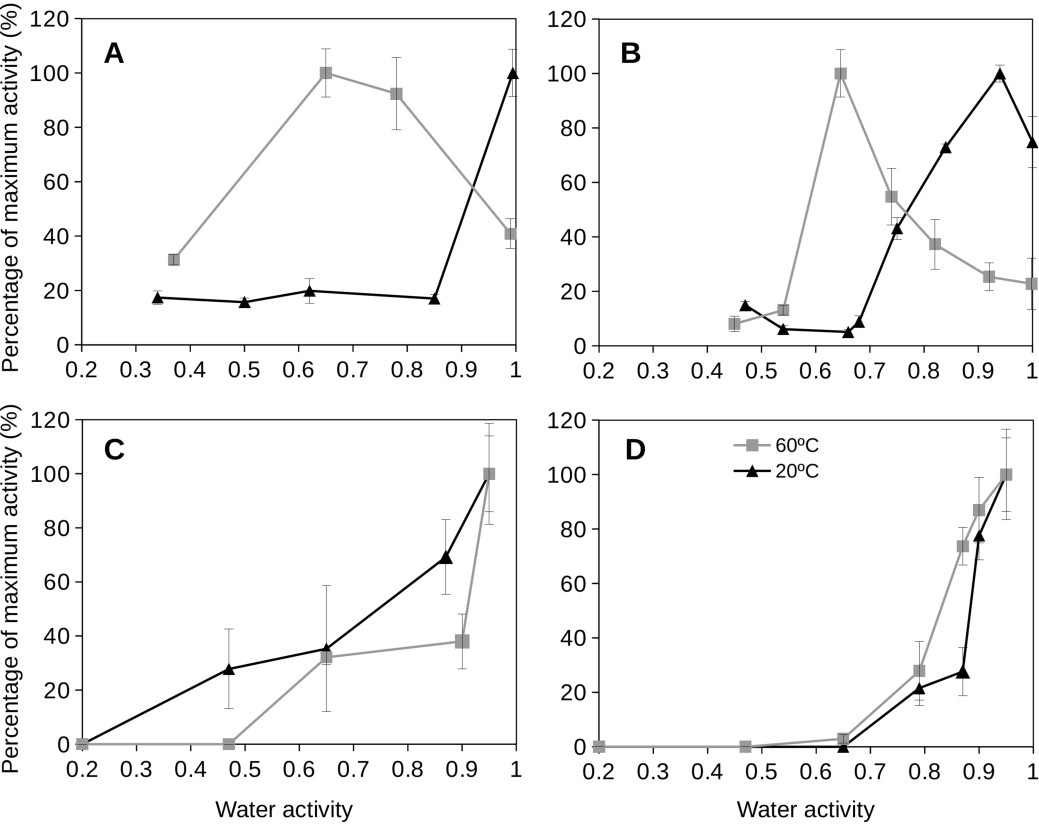

**Figure 5 Extracellular enzyme activities at different temperatures as a function of water activity for two distinctive soils.** Extracellular enzyme glucosidase (A and C) and protease (B and D) activities at different temperatures (20 °C, blue triangles, and 60 °C, red squares) as a function of water activity ($a_w$) for two distinctive soils, one frequently exposed to heat and dryness (Southern Spain; (A) and (B)), and the second one from a wet and moderate temperature soil (Northern Spain; (C) and (D)). Error bars represent one standard deviation.

activities (from dryness to wet conditions). Experimental estimations of extracellular enzyme activity from natural soil samples showed different responses depending on both temperature and water activity (Fig. 5). For a soil frequently exposed to high temperatures (Southern Spain soil), the optimum values of enzyme activity at the mesophilic condition were observed at high $a_w$ values (at 0.95). This soil at the thermophilic condition showed optimum enzyme activity values at $a_w$ 0.65 (Fig. 5), with reduced activities at $a_w$ 1 and undetectable activities at very low values of $a_w$ (<0.3–0.4).

When analyzing the results of extracellular enzyme activities at different temperatures and water activities for a soil rarely exposed to high temperatures and in a wet environment (Northern Spain), the results were quite different (Fig. 5). In this soil, the optimum extracellular enzyme activities (both for mesophilic and thermophilic acitivities) were observed at high $a_w$ (around 1) and the enzyme activity decreased when reducing water activities. These results showed a clear differentiation of the studied soils according to their extracellular enzyme activity which is strongly dependent on temperature and water activity.

## DISCUSSION

Temperature and water availability are two major environmental factors directly affecting microbial community structure and its functioning. However, our understanding of the effect of temperature and water availability on the activity of extracellular enzymes in the environment is very limited (*Gonzalez, Portillo & Piñeiro-Vidal, 2015*; *Wallenstein & Weintraub, 2008*) and this represents a serious handicap to evaluate future models and potential scenarios of C fluxes at local and global levels (*Santana, Gonzalez & Garbeva, 2015*). Previous publications (*Gonzalez, Portillo & Piñeiro-Vidal, 2015*; *Wallenstein & Weintraub, 2008*) have emphasized the importance of temperature on enzyme activity and the fact that soil enzyme activities are typically only measured at a single temperature and often far from the field site temperature. Besides, previous estimates of EEA correspond exclusively to relatively high water availability (*Stark & Firestone, 1995*; *Mathlouthi, 2001*; *Steinweg, Dukes & Wallenstein, 2012*; *Borowik & Wyszkowska, 2016*) and desiccation conditions, particularly in arid ecosystems, remain to be studied (*Moxley et al., 2019*). The present study contributes to our understanding of the microbial extracellular enzyme activity as a function of temperature and water activity by proposing a novel methodological strategy approaching enzyme activity under the effect of these environmental parameters (temperature and water availability) in a highly heterogenous particulate support media (i.e., soil).

The proposed procedure consists on reducing the water availability by freeze-drying the studied samples so that the enzyme activity can be evaluated under the desired conditions of water activity. Although freeze-drying processes have been reported to potentially cause moderate loss of activity to some enzymes (*Valaskova & Baldrian, 2005*), our results showed minimum effects on soil extracellular enzyme activity at high water activity values in solution and solid assays (Fig. 3). Adapting the protocols of enzyme assays to this important step allows to approach realistic measurements of the role of extracellular enzymes under these, so far, scarcely studied conditions. Besides, variations of temperature and water content are frequently observed at the upper layers of natural soils. Using this simple protocol that gap can be filled to understand how the extracellular enzymes function during the periods that are exposed to high temperature and low water availability. So far, estimations performed in the environment have been commonly carried out in solution which provides a potential measure of extracellular enzyme activity under wet, high water content, conditions. Nevertheless, upper soil layers can frequently get hot and dry mostly in arid systems.

When measuring microbial extracellular enzyme activities in soils one should be aware that soils are highly heterogeneous, particulate environments. In this type of media, proteins, such as extracellular enzymes, tend to adher to soil particles (*Kolman et al., 2014*). Enzyme adsorption to particles is an immobilization process involving diverse physical interactions which could result in a reduction of enzyme activity (*Allison & Treseder, 2008*; *Datta et al., 2017*; *Rodrigues et al., 2013*) as a consequence of possible conformational changes in the enzymes. Besides, it is well known that immobilized enzymes are subject to substrate diffusion limitations (*Datta et al., 2017*; *Duran & Esposito, 2000*; *Tecon & Or, 2017*).

These issues could also occur under certain conditions in soils resulting in changes of enzyme assay estimates at decreasing water activity in natural soils under desiccation. The present study shows that temperature and water activity influence microbial EEA in soils.

Although a Proteobacterium, *P. aeruginosa*, presents optimum values of extracellular enzyme activity under high water content, we have shown that a desiccation-resistant species (i.e., *D. radiodurans*) (*Potts, 1994*; *Slade & Radman, 2011*) helds optimum behavior of its extracellular protease and phosphatase activities under reduced water availability (0.55 $a_w$). On the other hand, *D. radiodurans* glucosidase activity peaked at $a_w$ 1 which could suggest a different behavior of enzymes targetting polysaccharides vs enzymes related to the N and P cycles in this organism.

The behavior of the extracellular enzymes from a thermophilic bacterium (i.e., *P. thermoglucosidasius*) was studied. This is a Firmicutes representative of thermophilic soil bacteria generally detected as viable cells in soils even at temperate and cool zones (*Marchant et al., 2002*; *Portillo, Santana & Gonzalez, 2012*; *Wong et al., 2015*). The extracellular enzymes from *P. thermoglucosidasius* presented singular peaks of maximum activity under low water activity ($a_w$ 0.5–0.8) corresponding to quite dried soils. During hot (and high solar exposition) periods evaporation increases as soil warms up, the water content in soils decreases, then it is reasonable that soil thermophilic bacteria show high activity rates under those extreme temperature and desiccation conditions at upper soil layers.

Previous reports (*Grant, 2004*; *Stevenson et al., 2015*) have showed the drastic negative effect that reduced water availability can have on bacterial growth. In fact, the lowest water activity values that are accepted to support bacterial life has been reported at $a_w$ 0.75 for halophilic bacteria (*Grant, 2004*) and *Escherichia coli* barely survives at water activity at, or below, 0.95. It has been reported that the lowest limit of water activity that enables cell division is around 0.605 for the xerophilic fungi *Xeromyces bisporus* (*Stevenson et al., 2015*). Herein, we have shown that extracellular enzymes (e.g., *Deinococcus* and *Parageobacillus*) show activity and function optimally below the limit of water activity for microbial growth.

We also tested whether extracellular enzymes from microbial communities at natural soils also present different behavior depending on their soil environment. We evaluated two soils from different locations, one frequently exposed to heat and dryness (Southern Spain) and the second one rarely exposed to these conditions (Northern Spain). Enzymes from the mesophilic microorganisms, both from the Southern and Northern Spanish soils, showed optimum activities at high water activities ($a_w$ > 0.95), the extracellular activity from the thermophilic microbial population (at 60 °C) at a heat and dryness exposed soil (Southern Spain) presented optimum (maximum) enzymatic activities at reduced water activities ($a_w$ 0.65). The extracellular enzyme activity from thermophiles at a cool soil (Northern Spain) showed the peak of maximum enzyme activity at high water activity similarly to their mesophilic counterparts. Mesophiles thrive under low temperature conditions, a more permissive scenario to conserve water that the high temperature extremes required by soil thermophilic activity. This study represents a first report suggesting a potential

capability of natural microbial communities to adapt to the environmental conditions existing in their niches.

## CONCLUSIONS

How extracellular enzymes function in highly heterogeneous particulate media, such as soils, is a topic that requires further investigation. Temperature and water activity are two parameters that show differential and decisive influence on extracellular enzyme activities of soil microbial communities. Our results show that the extracellular enzymes from microorganisms inhabiting, for example, arid and semi-arid soils present relatively high capability to decompose complex organic matter under conditions of water scarcity than previously assumed. These results complement current attempts to understand microbial biogeography by suggesting that bacterial physiology and, likely, distribution can be influenced by environmental factors such as temperature and water content.

### Funding

This study was supported by funding through projects from the Spanish Ministry of Economy and Competitiveness (CGL2014-58762-P) and the Regional Government of Andalusia (RNM2529 and BIO288). These projects have been cofunded by FEDER funds. The funders had no role in study design, data collection and analysis, decision to publish, or preparation of the manuscript.

### Grant Disclosures

The following grant information was disclosed by the authors:
Spanish Ministry of Economy and Competitiveness: CGL2014-58762-P.
Regional Government of Andalusia: RNM2529 and BIO288.
FEDER Funds.

### Competing Interests

The authors declare that they have no competing interests.

### Author Contributions

- Enrique J. Gomez conceived and designed the experiments, performed the experiments, analyzed the data, authored or reviewed drafts of the paper, and approved the final draft.
- Jose A. Delgado performed the experiments, analyzed the data, authored or reviewed drafts of the paper, and approved the final draft.
- Juan M. Gonzalez conceived and designed the experiments, analyzed the data, prepared figures and/or tables, authored or reviewed drafts of the paper, and approved the final draft.

### Data Availability

Raw data are available as a Supplemental File.

## Supplemental Information

Supplemental information for this article can be found online at http://dx.doi.org/10.7717/peerj.10994#supplemental-information.

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
