# Peer review of "Influence of water availability and temperature on estimates of microbial extracellular enzyme activity"

_PeerJ, doi:10.7717/peerj.10994_

## Round 0.1 · original submission · Major Revisions

Hello. You have two reviewers go over your MS, and one is quite critical to some of the methodology, which I believe might be due to the fact that the description is unclear, for example: Is pH 11 that of the buffer of that of the treatment? In any case, both reviewers mentioned soil characteristics such as granulometry/homogeneity, so you must address these concerns in your answer

Reviewer 1 ·

Basic reporting

Generally well written, and very well referenced.

Experimental design

See below

Validity of the findings

General Point
The authors appear to have ignored the fact that the three enzymes chosen are all hydrolases (requiring water for their hydrolytic reactions). Reductions in catalytic rates may therefore be the consequence of substrate limitations (at some point), accepting that [H2O] substrate is 55M at Aw=1. This issue could be addressed by expanding the enzyme range to include some non-hydrolytic enzymes.

Methodology
Line 169 onward. Having spent some time explaining that soils are highly heterogeneous, and soil microbes are found in a wide range of microenvironments in fact, most in biofilm assemblages within the interstices of soil conglomerates and particles), the authors then use a protocol which involves the supplementation of soils with pure microbial cultures. These cells are likely to mostly be adsorbed to soil particle surfaces, will not be in biofilms, and are unlikely to represent the ‘native state’.
Given this concern, the authors would be advised to provide more basic information on the mineralogy of their two samples (sand/clay/silt content; particle size range).

Line 199. 2M phosphate buffer – this is an extraordinarily high buffer concentration, and would certainly affect the ionic environment of the enzyme.

Line 209 – specify the substrates (for those readers who don’t have instant access to Gonzalez et al 2015).

Line 207 onward. This is a complicated protocol, with many points at which the fluorogenic substrates (which are known to be reasonably labile) might be autolysed non-enzymatically. I don’t have the references to hand, but I do remember old literature which shows that some soils (clays?) are quite potent catalytic surfaces. I just don’t see the evidence for the negative controls.
I am particularly concerned about the possible abiotic effects on the pH adjustment (pH 11, line 225), since many of the chromogenic and fluorigenic substrates are VERY subject to alkaline hydrolysis.
I also have some concerns about the freeze-drying process for establishing the ‘desired’ water activity. It is not clear how the ‘desired’ water activity was reached – was freeze-drying terminated at intervals, the sample thawed and tested with the Rotronic probe, then f-d some more? Aw measurements with the probe could only be done on thawed samples (at constant temperature).

Results
I am struggling to understand, mechanistically, how enzyme activities can reduce at high Aw levels (Fig 3). These are all hydrolytic enzymes, so water is involved in the enzymic reactions, and reduced rates at lower aw levels are therefore entirely to be expected (apart from the effects of low Aw on the enzyme structures).
I can’t help feeling that the reducing ‘activities’ at high Aw levels (Figs 3A, 3B) are the product of some uncharacterised artefact. The authors attempt to interpret these results in an ecological context (hot dry southern Spain vs cool wet northern Spain: that the microbial communities, through their enzymes, adjust their physiology to the environment (differential adaptation patterns) are not wholly convincing. At the very least, the authors should include macro/microenvironmental data from the locations to support these statements.
There are also some other data that make me very worried that the data are largely artifactual – for example, Fig 3C (pale squares). I find it quite inexplicable that enzyme activity would increase so dramatically from Aw 0.9 to Aw 0.95 (which represents a rather small change in water availability). Basically, I think that there is more going on here than can be explained.

I would want to see substantive supporting data before I could accept this rather ‘circumstantial’ ecological correlation. I just don’t think that the data are strong enough, particularly in light of the complexity and uncertainty of the assay systems (see comments above), for such broad fundamental conclusions.

Minor issues
Line 308 and elsewhere – the term ‘soggy’ is a colloquialism!

Additional comments

Despite the fact that there could be something very interesting here, my view is that there are just too many concerns and unanswered issues in this piece of work. Even if the assay results are accurate and valid, the attempt to place them in a macroecological and evolutionary context is just a 'step too far'.

·

Basic reporting

Overall this manuscript is well-written and concise. I listed a few sections below that you may consider revising.

Title: It is a bit strange/confusing that it has a period in the middle.

Lines 111-115: Awkward sentence.

Lines 151-154: Awkward sentence.

Lines 384-387: Awkward sentence.

Line 389: You only list one study here, but mention previous "reports".

Figure 1: Can you indicate which point corresponds to which enzyme and make the error bars more clear? Additionally, please indicate that the axes are in the log scale and add the correlation coefficient to the plot.

Experimental design

This study presents a new approach to measuring extracellular enzyme activity in soil and explores the effect of water availability on extracellular enzyme activity. Soil moisture level is integral part of microbial ecosystem response, yet the assays for measuring extracellular enzyme activity rely on an aqueous solution. Thus, this study is a very exciting contribution to the field of soil microbial ecology. However, I have a few concerns/questions about the methodology that should be clarified or explained. Particularly, how do you know that the measured enzyme activity is not just a reflection of how well you mixed the substrate with the soil? I understand the purpose of the procedure is to better replicate in situ conditions, but with such short incubation time, it seems like this enzyme activity would be highly based on mixing success.

Other minor comments:
Lines 170-173: I’m assuming these bacterial strains were isolated from soil at these sites, but I didn’t see that explicitly stated in the manuscript.

Lines 177-179: What was the water activity or capacity of the soil when it was autoclaved? Was the water activity different for the different soils/treatments? The amount of water can affect how well the soil was autoclaved.

Lines 207-234: Can you add a photo or figure detailing this procedure? I don’t quite understand how you added the substrate in a non-aqueous solution, which is important since this is a major focus of your manuscript.

Lines 220-221: Can you show an example kinetic curve from this study? I think it may be helpful since this is a new method and readers might have questions about incubation times and substrate saturation when conducting their own experiments and to show that you met these criteria as well.

Validity of the findings

With additional clarification of the methods, I think the findings will be better supported. However, overall I found the conclusions reasonable based on the evidence provided. I have also listed some other minor concerns below.

Other minor comments:
- There seems to be an underlying assumption throughout the manuscript that optimum growth temperature, optimum temperature of enzyme activity, and mean environmental temperature are the same. However, this is likely not true (see, Prentice et al., Biochemistry 2020 and Alster et al. 2018 Glob. Chang. Biol.). I would make sure to clarify this throughout the text.

Line 256: If the results are equivalent, what is the advantage of using this new method?

Lines 366-368: This is a strong claim. Did you provide statistical evidence for each of these factors in the manuscript?

---

## Round 0.2 · Minor Revisions

Hello. As you can see the reviewer has further comments to your MS. Could you please address them in a resubmission. Thank you

·

Basic reporting

The authors did a great job revising this manuscript and incorporating the relevant comments from myself and the other reviewer.

Minor comments:
1. Lines 173-178: I appreciate that you added this sentence, but it is very long and therefore confusing to read.
2. Line 198: Is there a typo here? Do you mean “scheme” instead of “schee”?
3. Figure 3: Are the values in the log scale or are the axes? The way this has been changed from the prior draft is confusing.
4. Line 143: Change “on” to “of”

Experimental design

The addition of Figure 1 was very helpful in clarifying the methods. Thank you for adding that. However, I still have some questions about the incubation times. In particular:

1. Lines 215-222 and 253-255: Was the total incubation time only 10 minutes for all of the different temperatures and moistures? It would be helpful if you could clarify this specifically in the methods. That seems on the shorter side for the lower temperature incubations.
2. How do you know that when you did the drying step you didn’t remove some of the florescent substrate? Maybe you explained this somewhere and I didn’t understand, but I think this is important to clearly demonstrate or note. I think that is maybe what you are showing in Fig. 2, but you only show the high moisture level.

Validity of the findings

The authors narrowed the scope of the discussion from its prior version and it now better represents the findings.

Additional comments

I would like to see how the authors address the substrate saturation and incubation time questions, but overall this seems to be an exciting new method that I would be eager to try in my own experiments. Also, interesting result of peak EEA at lower moisture levels for some of the species. Very relevant to the field - Nice job!

---

## Round 0.3 · Minor Revisions

Hello. I would like you to please make the following editorial revisions which I hope will improve your MS
Thank you very much

line 19 “While this process occur on soil particles, currently it is typically ... "
Line 25 "dry" instead of "dried"
Line 29 extremely low
Line 30 these improved estimates ... will contribute
Line 34 critical instead of decisive
Line 43 Macro-organisms
line 118. add "thus" before "excluding"
line 120 enzyme activity in soil has been measured at
line 122 delete "but these values" then "also representing"
Line 125 "Currently" instead of "so far"
line 209 "close" not "closed"
line 213 include details for : "tightly closed"
line 252. incubation
line 273 delete "from the use of"
line 277 : proteobacterium
line 278. When the water activity was reduced
line 281 was reduced .. remained
line 286-294 pasrt tense was evaluated ... were slightly ... was analyzed ... was constant ... drastically dropped ...was sharply
line 306 delete "attempting"
line 309 "resposes" instead of "functional models"
line 311 "the mesophilic condition"
line 312 "thermophilic condition"
line 318. "were quite different"
line 335 "and dessication conditions, particularly in arid ecosystems"
line 396. replace "the next .. evaluate if" with. "We also tested whether"

---

## Round 0.4 · accepted · Accept

Thanks for answering my comments.